# A Tragedy of Errors: The State of Psychedelic Research in the Treatment of Alcohol Use Disorder

**DOI:** 10.3390/brainsci15111190

**Published:** 2025-11-04

**Authors:** A. Benjamin Srivastava, Mark S. Gold

**Affiliations:** 1Department of Psychiatry, Columbia University Irving Medical Center, New York, NY 10032, USA; 2Department of Psychiatry, Washington University School of Medicine, Saint Louis, MO 63110, USA; drmarksgold@gmail.com

**Keywords:** alcohol use disorder, psychedelic, psilocybin, alcoholics anonymous, dorsolateral prefrontal cortex, cognitive control

## Abstract

The past two decades have seen the reemergence of research investigating the therapeutic potential of psychedelic drugs across neuropsychiatric illnesses. One condition, alcohol use disorder (AUD), is of relevance given the broad public health implications and both limited effectiveness and attrition associated with currently available treatments. While emerging research has suggested that the benefits of psychedelic drugs in the treatment of AUD may be considerable, several fundamental aspects of this work limit the conclusions that can be drawn. These limitations include those that apply to research involving psychedelics generally—including functional unblinding and the role and definition of “psychedelic assisted psychotherapy” and some unique to AUD, including the nature of the mystical experience and how it relates to the “spiritual experience” as described in the literature of Alcoholics Anonymous (AA), of which the history of psychedelic research in AUD is closely intertwined. Additionally, current mechanistic neuroimaging studies examining the therapeutic effects of psychedelics in AUD are limited by design and do not directly interrogate the cognitive and circuit-level processes likely underlying treatment response. This review describes these limitations in detail by bridging historical, conceptual, and mechanistic aspects of psychedelic research in AUD and offers suggestions for future studies, the results of which may more clearly specify the role and utility of psychedelic drugs in the treatment of AUD.

## 1. Introduction

Alcohol use disorder (AUD) is a significant public health problem with significant morbidity and mortality, contributing to over 178,000 deaths annually in the United States [1,2,3]. Treatment guidelines recommend medications for alcohol use disorder (MAUDs), and currently, three agents are FDA approved for the treatment of AUD: naltrexone, disulfiram, and acamprosate [4]. Recommendations notwithstanding, recent meta-analyses have shown small–moderate effect sizes with these medications in blinded studies, and adherence is generally poor [5,6]. Therefore, new interventions are desperately needed. Psychedelic drugs that are agonists at the serotonin 2A receptor (5HT2A) [7], namely, psilocybin and lysergic acid dimethylamine (LSD), have garnered substantial interest as novel therapeutics for a number of psychiatric illnesses, including AUD [8,9,10]. While recent studies have demonstrated positive results [11,12], we view these findings with extreme skepticism because of fundamental conceptual errors in experimental design and scientific reasoning. Randomized controlled trials (RCTs) investigating psychedelic and entactogenic agents (e.g., 3,4-methylenedioxymethamphetamine [MDMA]) [7,13] face several recurring challenges, including inadequate blinding (with 80–90% of participants correctly guessing their assignment) [8,14,15,16,17,18], limited consistency and scalability of “psychedelic-assisted psychotherapy” [19,20], and serious ethical concerns [21]. The first two issues are without question present in the extant AUD literature and will be addressed specifically as they relate to AUD. Additionally, current literature on therapeutic mechanisms of psychedelics in AUD is characterized by significant methodological limitations. In this review, we will discuss these conceptual issues ensconced in current psychedelic research and suggest avenues for improvement. The terms “alcoholic” and “alcoholism” invoke stigma and discrimination [22]; however, in the older literature, these terms were acceptable and will be used in this article interchangeably with AUD when appropriate (Figure 1).

## 2. Psychedelics and Alcohol Use Disorder: The Historical Concept

To fully understand the pitfalls of current psychedelic research in AUD, an accurate understanding of the historical context in which the ideas of using psychedelics for therapeutic purposes developed is indispensable, and a thorough grasp of this history is incomplete without discussing its indelible ties to Alcoholics Anonymous (AA). AA is a self-governed, mutual support organization founded in 1935, aimed at achieving complete abstinence from alcohol through undergoing a “spiritual experience” [23,24,25,26]. The history of AA itself has been extensively documented and discussed (see [27,28]), but in short, William Griffith Wilson (“Bill W.”, or “Bill”), a previously successful New York stockbroker whose personal and professional life was in ruins because of excessive drinking, acquired means of recovery through engaging in practices of the Oxford Group, an Evangelical Christian group centered on achieving a spiritual transformation through a process of self-appraisal, restitution, and service. During his third hospitalization for alcohol withdrawal, he experienced a transformative experience, described by Carl Jung as a “vital spiritual experience” defined as a phenomenon characterized by “huge emotional displacements and rearrangements… ideas, emotions, and attitudes… are suddenly cast to one side, and a completely new set of conceptions and motives begin to dominate them” [25,29]. Following this experience, Bill remained sober for the rest of his life [29]. Of note, Bill was treated with *Atropa belladonna*, a compound with anticholinergic effects [30] that has been speculated to have been causally involved in his experience [29]. Regardless, Bill then realized that the practices of the Oxford Group could lead to this necessary spiritual experience, later defined as a “personality change necessary to bring about recovery from alcoholism” [25], and he dually understood that he needed to transmit these ideas to other alcoholics. After six months of failing to persuade any alcoholics of his ideas of recovery, during a failed business venture in Akron, OH, he was introduced to Dr. Robert Holbrook Smith (“Dr. Bob”), a previously well-respected surgeon whose years of drinking had similarly caused immense psychosocial, professional, and interpersonal dysfunction. Dr. Bob became sold on Bill’s ideas and remained sober, and over the next four years, they were able to expand their fellowship, both in Akron and New York, culminating in a formalized plan of recovery, termed the Twelve Steps. Deeply influenced by William James’s *The Varieties of Religious Experience* [31], Bill, Dr. Bob, and others believed that most spiritual experiences were not of the sudden kind Bill had experienced, but occurred slowly over time, working through the Twelve Steps. In 1939, they published *Alcoholics Anonymous* (“The Big Book”), the primary text of AA outlining the program of recovery, from which the fellowship is named. AA has had an immeasurable influence: AA reports 2 million members worldwide, the Big Book was designated as one of 88 books that “shaped America” in the United States Library of Congress, and in 1951, AA was awarded the Lasker Award from the American Public Health Association [23].

According to the Alcoholics Anonymous literature, a necessary condition for achieving the spiritual experience is “hitting bottom,” which is embodied in the first step: “We admitted we were powerless over alcohol; that our lives had become unmanageable” [25]. At this point, one admits “complete defeat”—that he or she lacks the ability to control how much alcohol he or she drinks (“powerlessness”) and that, as a result, experiences profound negative consequences (“unmanageability”) [25,26]. While this step is a prerequisite for recovery, it is generally not, according to AA literature, sufficient for achieving the necessary spiritual experience: the individual must work through the remaining eleven Steps to attain sustained abstinence from alcohol [25,26]. Dr. Harry Tiebout, a psychiatrist and early observer of AA, expanded on these ideas in a seminal 1953 paper [32], in which he described that for abstinence to be achieved, “hitting bottom” must be followed by “surrender” and ultimately “acceptance”, where “relaxation ensues with freedom from strain and conflict”. This state of acceptance, which may be accomplished through the Twelve Steps, is tantamount to the necessary spiritual experience. In a survey of members of AA, the majority of those endorsing spiritual experiences reported that spiritual awakening occurred over time while working through the Twelve Steps [33], further supporting the primacy of the process-oriented nature of recovery in AA.

In the 1950s, Drs. Humphrey Osmond and Abram Hoffer, psychiatrists at the University of Saskatchewan, theorized that “hitting bottom” as described in the AA literature could be produced in alcoholic patients through a “controlled” form of delirium tremens, the most severe form of alcohol withdrawal characterized by profound disorientation, autonomic instability, and psychosis [34,35,36]. The psychotogenic properties of LSD, which was synthesized by Albert Hoffman at Sandoz Laboratories in Switzerland, had already been described [9,37], and Osmond and Hoffer speculated that the LSD toxidrome may approximate the experience of delirium tremens [34]. Thus, Osmond and Hoffer began testing LSD on their patients with AUD in uncontrolled, observational studies [34,36], finding generally positive results, often with significant improvements in abstinence from alcohol. One unexpected but fundamental realization was that patients reported what Osmond came to define as a “psychedelic” experience following administration of LSD—a sensory distortion followed by an “enrichment of the mind and enlargement of vision”, which they saw as complementary to formal, structured treatment; invariably a therapist was involved in the administration of LSD, often ingesting LSD himself or herself during the session [34,36]. This sporadic dosing became conventionally known as “psychedelic assisted psychotherapy” [9,19]. We note that contemporaneously in Europe, smaller doses of psychedelics, usually LSD, were administered over multiple sessions, termed “psycholytic therapy” [9,19]. Ultimately, Osmond and Hoffer described the therapeutic effects of LSD as leading to ego deflation, improved self-insight, and value reorientation [34,36], paralleling the “hitting bottom” leading to surrender and acceptance trajectory described by Tiebout.

One observation from Osmond and Hoffer, though inconsistently supported by data at that time, was that LSD may be useful for alcoholics who either refused to engage in or could not achieve sobriety through AA despite attempting to do so because they viewed “spirituality” as a barrier [34,36]. Bill W. was introduced to the findings of Osmond and Hoffer and was at first skeptical of giving medications with psychoactive effects to alcoholics, but upon reflection on his own spiritual experience, he became intrigued with some of the positive findings of previously refractory patients achieving success in AA, and he began to see a therapeutic potential in LSD [29,38]. Bill famously self-administered LSD, attempting to replicate the spiritual experience he achieved in 1935 [29,38]. Being influenced by James’ idea of “religiomania” as a “radical remedy for dipsomania” [31], Bill understood the absolute necessity of enhancing the spiritual experience for sustained sobriety and saw LSD as an agent that could facilitate this process [29]. While much of this history includes practices today that would not be considered acceptable or ethical in contemporary clinical research, and data presented are often inconsistent, conflicting, and with small sample sizes, two major themes emerge: (1) the history of Alcoholics Anonymous and the therapeutic use of psychedelics in the treatment of AUD are closely intertwined and (2) the therapeutic effect of psychedelics in AUD is purported to occur through the facilitation of a spiritual experience. We argue that these themes must be given paramount consideration in translational research investigating psychedelics in the treatment of AUD. We recognize that the AA vernacular is not derived from standard clinical or scientific language; however, we will utilize this language to maintain conceptual continuity and, when appropriate, relate it to normative clinical and scientific terminology. Further, we note that many studies have described the subjective experiences of psychedelics as “mystical” or “mystical-type” experiences; for the purposes of this article, the terms “spiritual experience” and “mystical experience” will be used interchangeably [39].

## 3. Psychedelics and Alcohol Use Disorder: The Evidence

### 3.1. Clinical Outcomes

The study of the therapeutic effects of psychedelics was halted in the 1970s due to the passage of the Controlled Substances Act [37,40], but as discussed, recent years have seen a renewed interest in research into psychedelics for therapeutic purposes. See Table 1 for an overview of recently published studies examining the efficacy of psilocybin and LSD in the treatment of AUD. An often-cited 2012 meta-analysis examining six randomized controlled trials [41] is used as justification for current and future translational research in AUD. In this meta-analysis, Krebs et al. found that a single dose of LSD conferred benefit for “alcohol misuse” (OR, 1.96; 95% CI, 1.36–2.84; *p* = 0.0003) [41]. However, several fundamental concerns limit the interpretability of this meta-analysis. The comparator interventions were not standardized and included stimulants, lower doses of LSD, treatment as usual, and “Sit[ting] alone and writ[ing] for 3 h”; outcome measures were nebulous and inconsistent, including non-standard measures, such as “adjustment” and “drinking behavior”’; and psychotherapy treatment lengths and modalities varied and included community based, group, and individual therapy [41]. Nevertheless, based on both the positive findings and the need to standardized dosing and therapy, Bogenschutz and colleagues [12] conducted a small, open-label study investigating the effects of psilocybin in AUD, enrolling 10 participants in a 14-week study with weekly motivational enhancement therapy (MET) sessions with supervised psilocybin dosing sessions after 4 and 8 weeks. Preparation and debriefing sessions occurring to prior to and after psilocybin dosing, respectively. Notably, two therapists delivered the psychotherapy: one therapist was responsible for the supervision during psilocybin dosing preparation and debriefing, and the other was responsible for MET. The primary outcome was the percentage of heavy drinking days, an accepted outcome in clinical trials used by the FDA for granting approval for medications for AUD (with no HDD being the accepted primary endpoint) [42,43], with a heavy drinking day (HDD) defined as >4 drinks/day for men and >3 drinks/day for women [42,43]. Bogenschutz and colleagues found that participants experienced a reduction in PHDD over the course of treatment. Given that this was an uncontrolled pilot study, these results were considered preliminary and hypothesis-generating.

With the intervention and outcome measures now established, Bogenschutz and colleagues [11] conducted a follow-up blinded RCT in which participants with AUD were assigned to receive either psilocybin (n = 49) or diphenhydramine (n = 46), with all participants undergoing 12 weeks of a tailored psychotherapy featuring components of MET and cognitive behavioral therapy (CBT), along with supervised dosing sessions occurring after four and eight weeks, including preparation and integration sessions. Diphenhydramine was selected as an active placebo, given that its subjective (anticholinergic) effects (sedation, confusion, impairments in motor functioning) could potentially improve blinding success [44]. Participants were followed for a total of 36 weeks, with the primary outcome defined as PHDD. While pre-dosing (weeks 1–4) reductions were similar between the two groups, between weeks 5 and 36, participants randomized to psilocybin had a 13.9% lower absolute PHDD than those receiving diphenhydramine. For psilocybin, the mean [SD] was 9.71 [26.21], and for diphenhydramine, the mean was 23.57 [26.21], with a mean difference of 13.86; 95% CI, 3.00–24.72; and Hedges g = 0.52; *p* = 0.01 (medium effect size). Notably, when examining total abstinence, there was no difference between the two groups between weeks 5 and 36 (8.9% for diphenhydramine, 22.9% for psilocybin; OR 3.05, 95% CI, 0.89–10.40); however, between weeks 33 and 36, participants randomized to psilocybin were more likely to be abstinent than those randomized to diphenhydramine (47.9% vs. 24.4%; OR 2.84, 95% CI, 1.17–6.89). Of note, 93.6% and 94.7% of participants correctly guessed their treatment assignment in the first and second dosing sessions, respectively. Moreover, 92.4% and 97.4% of therapists correctly guessed the treatment administered in the first and second sessions, respectively, echoing the “functional unblinding” [45] found in trials investigating psilocybin for major depressive disorder (MDD) [16,18] and MDMA for post-traumatic stress disorder (PTSD) [14,15]. These results were heralded as extremely promising [46], prompting additional trials investigating the use of psilocybin assisted psychotherapy in the treatment of AUD, the details of which are reviewed elsewhere [47]. However, this trial includes several fundamental limitations that significantly constrain its interpretability, including functional unblinding, the lack of further characterization of abstinence as an outcome, and the absence of any consideration of AA in the study design or analysis.

**Table 1 brainsci-15-01190-t001:** Summary of recent studies examining the efficacy of psychedelics in the treatment of AUD. LSD = lysergic acid diethylamide; PHDD = percentage of heavy drinking days; MD = mean difference.

Study	Design	n	Intervention	Comparator	Outcome	Key Results
Krebs & Johansen, 2012 [41]	Meta-analysis of 6 RCTs	536	LSD 200–800 mcg	Varied	Alcohol misuse	OR 1.96; 95% CI; 1.36–2.84
Bogenschutz et al., 2015 [12]	Open-label pilot	10	Psilocybin 0.3–0.4 mg/kg	None	PHDD	Reduction in PHDD (MD 22.4; 95% CI; 8.7–43.2)
Bogenschutz et al., 2022 [11]	RCT	95	Psilocybin 20–45 mg/kg	Diphenhydramine 50–100 mg	PHDD	Reduction in PHDD (MD 13.9%; 95% CI; 3.0–24.7)
Rieser et al., 2025 [48]	RCT	60	Psilocybin 25 mg	Mannitol	Abstinence (4 weeks)	No difference

### 3.2. Functional Unblinding

Much has been written on functional blinding [17,45,49,50,51], but in short, the conscious experience itself is likely necessary for any behavioral change; whether it could be replicated to adequate levels of blinding in a medication with a different pharmacologic profile misses the point. The issue of blinding was extensively debated in the 1950s [34,36], with several investigators contending that the subjective experience itself obviated the purpose of blinding. Further, blinding is just one factor (in addition to randomization, allocation concealment, completeness of follow up, etc.) that mitigates bias in trial designs [52]. A parallel may be drawn to another medication for AUD, disulfiram, which has been approved for the treatment of AUD since 1951 [53]. Disulfiram leads to the accumulation of acetaldehyde, a toxic product of alcohol metabolism, via the inhibition of aldehyde dehydrogenase [53]. Alcohol consumption while taking disulfiram leads to an extremely unpleasant reaction resulting from this acetaldehyde accumulation, and knowledge and fear of the reaction itself when taking disulfiram drives the motivation to not drink [53]. Blinded trials have not shown that disulfiram is more efficacious than placebo; its effectiveness has been principally demonstrated in open-label trials, likely because one needs to be aware of the possibility of the disulfiram-alcohol reaction to change behavior [53,54]. Nevertheless, the lack of efficacy in blinded RCTs does not negate the real-world effectiveness of disulfiram, and it remains the standard of care for patients seeking abstinence [4].

Accordingly, conducting randomized open-label trials comparing psilocybin with current FDA-approved treatments for AUD may be a more appropriate strategy, as recently suggested [55]. However, careful attention must be paid to trial design, participant selection, and outcome measures. Findings from a recent meta-analysis [56] generally support the combination of psychotherapy and pharmacotherapy, including CBT and other modalities, so the manualized therapy from the Bogenschutz et al. RCT [11,57] could still be utilized. Regarding the study endpoint, a key aspect of both the recent open-label study [12] and blinded RCT [11] investigating the therapeutic potential of psilocybin in the treatment of AUD is the primary outcome of PHDD. Historically, recovery, especially related to AA and AA-based treatment, was synonymous with complete abstinence, though more recent definitions encompass reduced drinking [43,58] based on substantial evidence that reductions in heavy drinking (i.e., harm reduction) can lead to improvement in quality of life and alcohol-related health outcomes [42,59]. Naltrexone, a mu-opioid receptor antagonist approved by the FDA for the treatment of AUD, is available in both oral and extended-release injectable formulations and is primarily used to facilitate a reduction in drinking [60,61]. A recent meta-analysis evaluating 1992 participants across 15 studies demonstrated that compared to placebo, naltrexone was associated with a reduction in PHDD, with a weighted mean difference (WMD) of −5.1 (−7.16 to −3.04) [6]. In the Bogenschutz et al. trial [11], the between-group (psilocybin vs. diphenhydramine) difference was 13.9%, though this finding could be explained in part by sample error, overestimating the true difference. To evaluate superiority, an ideal study should include a 12-week prospective, randomized, open-label trial design, in which participants with AUD would be assigned to receive either 50 mg of naltrexone daily or psilocybin, dosed at weeks 4 and 8, with all participants receiving 12 weeks of a manualized therapy for AUD. A blinded design including a daily placebo in the psilocybin group (blinding for naltrexone) and a control condition in the naltrexone group (active placebo, inactive placebo, subtherapeutic psilocybin dose [51]) is unlikely to enhance the study design and therefore unnecessary. For example, Carhart-Harris and colleagues [18] conducted an RCT comparing psilocybin to escitalopram for the treatment of MDD, where participants were assigned to receive either 6 weeks of escitalopram with two dosing sessions of 1 mg psilocybin (escitalopram arm) or 6 weeks of placebo with two dosing sessions of 25mg psilocybin (psilocybin arm), and >90% of participants correctly guessed their assigned group, thus leading to “functional unblinding”. Similar results regarding blinding integrity would likely be expected for any blinded study comparing psilocybin and naltrexone.

### 3.3. Abstinence as a Primary Outcome

While harm reduction is an important goal and successful strategy for many patients, certain patients, particularly those with severe AUD, are unsuccessful at moderating, even with treatment, and thus abstinence may be necessary [62,63]. Further, abstinence has been associated with improved quality of life and better psychosocial outcomes [64,65]. Incidentally, the inability to moderate is a hallmark of the alcoholic as described in AA and embodied in “powerlessness” in Step 1 [25,26]. In the Bogenschutz et al. RCT [11], there was no observed effect in abstinence over the course of the study, but when focusing on the final 3 weeks (weeks 33-36), participants randomized to psilocybin had a higher likelihood of being abstinent than those randomized to diphenhydramine, indicating a potential benefit for abstinence specifically over a longer treatment duration. Therefore, a prospective, randomized open-label trial comparing psilocybin with disulfiram over 36 weeks would provide greater clarity. The extended follow-up period appears critical for determining differences based on the finding that only during weeks 33–36 was a difference in abstinence between the two groups detected in the Bogenschutz et al. [11] trial. Further, in a recent RCT from Rieser and colleagues [48], participants with AUD following medically supervised withdrawal were randomized to either psilocybin or placebo (mannitol), with three integration visits over the course of four weeks. There was no significant difference in abstinence duration between the two groups after four weeks. While speculative, one study design issue may relate to the relatively short follow-up period, further emphasizing the need for an extended trial duration. Additionally, participant selection would likely need to be restricted to participants with severe AUD per the Diagnostic and Statistical Manual for Psychiatric Disorders, Fifth Edition (DSM-5) [66] criteria, seeking abstinence, and/or those who had failed harm reduction to most effectively evaluate an abstinence-related outcome.

### 3.4. Considering AA in Future Study Designs

As extensively discussed, the histories of AA and the use of psychedelics in the treatment of AUD are intimately linked: the idea of using LSD was derived from AA’s concept of “hitting bottom” [34], Bill W. realized that LSD may facilitate the spiritual experience in otherwise refractory patients [29,38], and in the initial studies, engagement in AA was often listed as an outcome measure [34,36]. Therefore, examining whether psilocybin can enhance or facilitate the effect of AA is prudent and may provide fundamental mechanistic insights into the therapeutic effects of psychedelics in AUD. Because AA is self-run and not usually considered treatment, evaluating the efficacy of AA has been challenging [67,68]. Twelve-Step Facilitation (TSF) is an evidence-based psychotherapy related to AA, based on the philosophy of the Twelve Steps, aimed at engaging the patient in AA. TSF has been rigorously studied [69,70], and high-quality meta-analyses have shown that, compared to other forms of psychotherapy, including CBT and MET, the two modalities used in the manualized therapy RCT by Bogenschutz and colleagues [11,12,57], TSF yields superior abstinence-related outcomes and lower healthcare costs [70,71]. Even when evaluating reductions in heavy drinking, AA/TSF performs as well as MET and CBT [70,71]. Further, mediation analyses have shown that TSF leads to greater participation in AA, which then leads to an improvement in abstinence-related drinking outcomes [70,72,73]. Studies examining the effectiveness of mechanisms through which AA may provide benefit [23] have shown that the strongest effects occur through adaptive social network changes (i.e., the AA “group”), as well as cognitive and behavioral strategies outlined in the 1975 book published by Alcoholics Anonymous World Services, Inc. *Living Sober* [74] that boost self-efficacy and reduce negative affect across all levels of AUD severity. However, AA members with more severe disease tend to derive particular benefit from spirituality and the spiritual experience [23,75], which is more aligned with AA’s original program of recovery described in *Alcoholics Anonymous* [25].

The success of AA/TSF notwithstanding, treatment failure remains considerable. For example, in Project MATCH, a large, multi-center prospective RCT that compared TSF with MET and CBT, 24% of participants assigned to TSF were abstinent at 1 year post-treatment, compared with 14% assigned to MET and 15% in CBT [70,76], indicating that even with the most efficacious treatment (TSF), 76% of participants were unable to stop drinking. Accordingly, examining whether psilocybin can augment the efficacy of AA/TSF represents a logical next step in this line of investigation. Because of the historical alignment between psychedelic therapy and AA and that AA/TSF outperforms MET and CBT, an optimal study would consist of an open-label trial of treatment-seeking participants meeting DSM-5 criteria for severe AUD, with all participants receiving manualized TSF for 12 weeks. One cohort would be randomized to receive psilocybin (ideally at weeks 4 and 8, allowing for a direct comparison to prior studies), and the other cohort would be randomized to receive TSF alone. Percentage of days abstinent (PDA) over 12 months would serve as the primary outcome measure, and PHDD could be an appropriate a secondary outcome measure. Because the fundamental nature of the trial would be to examine whether psilocybin enhances the efficacy of AA through the facilitation of a spiritual (or mystical) experience that otherwise might not occur, and this experience is by itself a conscious, realized phenomenon, irrespective of the cause, a control condition is not necessary and would likely confound the findings.

When evaluating whether the spiritual experience relates to a change in drinking, appropriate outcome measures are crucial. In the Bogenschutz et al. RCT [11], instruments aimed at characterizing the mystical experience include the Mystical Experience Questionnaire (MEQ-30), a questionnaire that attempts to qualify the mystical experience [77,78]; the NEO Personality Inventory, a validated instrument that measures quantifiable personality traits [79]; and the Schwartz Value Survey (SVS), which is used for assessing personal values (Conservation vs. Openness to Change and Self-Transcendence vs. Self Enhancement) [80,81]. Given that psilocybin has been shown to increase “Conservation” on the SVS [82], and “Conservation” and “Transcendence” are negatively associated with substance use behavior [83,84,85], in a secondary analysis, Gold et al. [86] hypothesized that psilocybin would increase these dimensions, which would then be associated with a reduction in drinking. While they found a treatment effect of “Conservation”, there was no effect on drinking outcomes. Similarly, there were no relationships found between any domains in the MEQ-30 and drinking outcomes. In a separate secondary analysis, Pagni and colleagues [87] found that psilocybin was associated with reductions in neuroticism as well as increases in extraversion and openness on the NEO personality inventory. Though decreases in impulsiveness, a subscale of neuroticism, were correlated with post-treatment drinks per day (the primary outcome, PHDD, is not reported nor is any abstinence-related outcome) across all participants, this correlation was not significant in participants randomized to psilocybin specifically.

These findings suggest *prima facie* that effects of psychedelics, either via the mystical experience or change in personality, are not associated with changes in drinking; however, several conceptual errors exist in the study design, limiting the interpretation of these findings. First, engagement in AA, a program centered on recovery through a spiritual experience, was not assessed in a structured way, nor was TSF included in the study protocol in the Bogenschutz et al. RCT [11,44,57]. This point is relevant because the original studies on LSD and AUD, in part, examined whether LSD could improve AA engagement through means of facilitating a spiritual experience [34,36]. While the manualized therapy used includes components that have been shown to effectuate behavior change [57], this form of therapy, unlike TSF [88], does not include components centered on spirituality, as it directly relates to recovery from AUD. Therefore, a trial using manualized therapy from Bogenschutz may be less than ideal for characterizing the psilocybin-facilitated spiritual experience. We acknowledge this point is to a considerable extent speculative; however, whether the facilitated spiritual/mystical experience differs as a function of treatment has never been experimentally delineated.

Second, measures used (NEO personality, SVS, MEQ-30) are not specific for AUD, have been mostly studied and validated in healthy volunteers, and may not capture the essence of the spiritual experience and subsequent personality change as they relate to recovery in AUD. A more appropriate outcome measure may be the Religious Background and Behavior (RBB) scale, an instrument designed for and validated in participants with AUD in Project MATCH, which directly measures a person’s spiritual practices and beliefs through treatment [89]. Mediation analyses have shown that RBB measures increase through treatment, and these increases in part mediate the effect of TSF [90]. Thus, RBB may better assess the spiritual/mystical experience as it relates to AUD recovery, particularly in the context of AA/TSF. Finally, the level of Alcoholics Anonymous engagement, irrespective of whether the manualized treatment administered is TSF, should be gauged. The primary scale used for the assessment of involvement in AA is the Alcoholics Anonymous Inventory (AAI), which has been shown to be associated with drinking-related measures [91]. In the Bogenschutz RCT [11], psilocybin was associated with abstinence only at weeks 33–36, long after formal psychotherapy treatment ended, though engagement in AA was not directly assessed. If this difference were attributable to enhanced AA engagement, it would constitute an important finding, underscoring the need for prospective trials combining TSF/AA-based approaches with psilocybin or other psychedelic drugs, with inclusion of the AAI.

## 4. Whither Neuroimaging?

Generally, hypothesized neuroscientific mechanisms for the therapeutic effects of psychedelics have focused on receptor-level targets, particularly the 5HT2AR [7,92]. While these pharmacodynamic properties are well validated, they are unlikely to provide a sufficient explanatory mechanistic framework for the therapeutic effects observed in clinical settings. A complementary approach involves examining large-scale brain network function and connectivity, with functional magnetic resonance imaging (fMRI) serving as the primary method for investigating these effects [7]. A subset of participants in the Bogenschutz et al. RCT [11] (psilocybin: n = 6, diphenhydramine: n = 5) underwent fMRI scans 3 days before and 2 days after dosing with psilocybin, which included viewing pictures of alcohol as well as positively, negatively, and neutrally valanced non-alcohol pictures. Pagni et al. [93], in analyzing these data, highlighted that psilocybin treatment was associated with increased alcohol cue reactivity in the dorsolateral prefrontal cortex (DLPFC) as well as other prefrontal cortical areas. The DLPFC is considered a locus of cognitive control, defined as processes that coordinate attention, goal maintenance, and a suppression of competing thoughts, feelings, and behaviors to optimize goal-directed behavior and counter automaticity [94], and cognitive control is considered a crucial component of recovery in AUD [95,96]. Pagni et al. suggest that this increased activation of DLPFC with psilocybin suggests facilitation of cognitive control processes involved in recovery in AUD. However, several major limitations exist. First, the sample size is far too small for group-level comparisons [97,98,99], making any reasonable inference untenable. Second, task fMRI measures the blood-oxygenation-level-dependent (BOLD) response to a time-locked stimulus [100], and in this study, the participants simply viewed pictures passively and rated cravings; no cognitive control process was directly probed. The increased activation patterns, even if observed in an appropriately powered study, could not be justifiably attributed to cognitive control vs. any other process that might engage DLPFC.

Other studies, however, offer evidence for mechanisms through which psychedelic drugs may facilitate behavior change in AUD. In a landmark study, Siegel and colleagues [101] examined the effects on brain network function and connectivity of both psilocybin 25 mg and methylphenidate 40 mg in healthy adults (n = 7) in a randomized, crossover study. What is notable about this study is that the authors leveraged precision functional mapping (PFM) techniques [100,102,103], wherein large amounts of data are acquired in individual subjects. This approach enables robust estimation of subject-specific connectivity and activation patterns, thereby permitting reliable inference, even in relatively small sample sizes. They examined both resting state functional connectivity (RSFC), a measure thought to reflect stable patterns of brain connectivity and changes that may index those related to experience-dependent plasticity [104], and a perceptual matching task. They found that compared with methylphenidate, psilocybin caused substantial cortical and subcortical connectivity disruptions, driven by network desynchronization, most noticeably in the default mode network (DMN), a network that is active when the brain is not engaged in a goal-directed task and widely thought to be involved in the creation and maintenance of predictive models of the external world that draws upon memory and experience as they relate to incoming sensory information [105,106,107,108], and these changes were reduced with task engagement. Further, whole-brain RSFC changes and MEQ-30 scores were strongly correlated (R^2^ = 0.81) on a session-by-session basis within each subject, providing strong evidence a circuit-level mechanism underlying the mystical experience.

Because of its methodological superiority, the study from Siegel et al. provides a crucial reference point for other studies utilizing fMRI in examining potential therapeutic effects of psychedelic drugs. Following this work, an attractive hypothesis would suggest that the spiritual experience facilitated by psychedelics in the treatment of AUD would reflect DMN desynchronization. As discussed, Gold et al. [86] demonstrated that MEQ-30 scores were not associated with changes in drinking behavior in AUD participants undergoing psilocybin treatment, though, as stated, the treatment frame (i.e., non-TSF-based manualized treatment) and MEQ-30 may be less than ideal for testing this hypothesis. Rather, future studies should include participants undergoing TSF and examine if DMN connectivity changes correlate with RBB scores.

Examining mechanistic work on AA itself may provide additional insight and avenues for future research. In a study conducted by Galanter et al. [109], members of AA with long-term (>2 years) abstinence and self-reported spiritual awakenings were asked to first either read prayers associated with AA (the Serenity and 3rd Step Prayers [25,26]) that involve acceptance and surrender, then think about the meaning of the prayers, and finally rate cravings for alcohol-related visual cues while undergoing fMRI scans. Control conditions included instructions to either passively view the alcohol-related visual cue or read an unrelated news clip and then think about the news clip while viewing the alcohol-related visual cue and subsequently rate alcohol cravings. Craving scores during the prayer condition were reduced, and prayer was associated with activity in the anterior middle frontal gyrus (MFG) area of the DLPFC, approximating area 9–46 [110]. What is notable about this task is that it employed a paradigm involving reappraisal, an emotion regulation strategy that involves the re-evaluation of the value of a salient stimulus [111,112]. Reappraisal is considered a form of cognitive control [111], and paradigms involving reappraisal have been used to examine how cravings for substance abuse are regulated in the Regulation of Craving (ROC) task. In this task, thinking about the long-term, negative consequences of substance use when viewing a drug or alcohol related cue is associated with a reduction in craving across substances, including alcohol [113,114,115,116,117,118]. Further, meta-analyses of reappraisal tasks (largely including emotion regulation as well as ROC) [119,120] and individual tasks using ROC in substance use disorders, including alcohol, have shown that regulation is associated with activity in a different DLPFC area, the posterior MFG, approximating area 8 [113,114,118]. Figure 2 visualizes this distinction on a single brain map [121,122], underscoring that “regulation” is not a unitary construct: ROC-based reappraisal and AA-related recovery processes engage spatially and functionally distinct DLPFC subregions.

These activation differences suggest potential mechanisms through which engagement in AA provides sustained and durable recovery. In the ROC task, the reappraisal strategy directly focuses on changing the value of the alcohol stimulus, and thinking through the consequences of drinking is a core component of both CBT for AUD [123] and Step 1 in AA/TSF [88]. In the AA task described above, the prayers relate to surrender and acceptance, without a direct focus on the alcohol stimulus itself. While both strategies lead to reductions in craving, they are associated with anatomically distinct areas within the DLPFC. The posterior MFG/area 8 is not well characterized and likely overlaps with several networks, including the DMN. Other networks potentially involved underlie language (language network) [124,125] and momentary task control (the fronto-parietal network) [126,127,128,129], the latter of which may be engaged, on a fast timescale, when confronted with alcohol “in the moment”. While this reappraisal strategy associated with posterior MFG/area 8 may be useful at times, it may not confer long-term benefit. In contrast, the anterior MFG/area 9–46 is a node of a network termed the Action Mode Network [127,128,129,130], a network proposed to be involved in the maintenance of goal-directed behaviors (i.e., staying abstinent) over a longer time course and may potentially be a locus of more durable recovery. A recent study examining the ROC task in participants with AUD undergoing CBT showed that neither the ability to regulate alcohol cravings through thinking about the negative consequences nor regulation-related posterior MFG/area 8 activity was found [118]. However, in this same dataset, connectivity changes between the anterior MFG/area 9–46 and anterior insula (AI), a region proposed to be involved in the regulation of craving and withdrawal [131,132], were associated, at a trend level, with reductions in drinking [133]. Additionally, a recent study using lesion network mapping [134,135] demonstrated that the anterior MFG/area 9–46 is part of an “addiction remission network”, further suggesting its involvement in long-term recovery-related processes.

A future study (NCT06349083) will recruit participants with AUD in residential treatment and randomly assign them to receive either 30 mg oral psilocybin or a placebo, along with three sessions of supportive therapy, with fMRI scans at baseline and two days after medication administration. The fMRI task will “characterize neural and subjective response to negative affective and alcohol visual stimuli”, though there is no mention of cognitive control processes directly probed. Therefore, attributing any DLPFC activation to improvement in cognitive control would be speculative. One of the primary outcome measures does include the Go/No-Go task with alcohol cues, administered four weeks after dosing, separate from the MRI, which probes response inhibition, a measure of cognitive control that may be associated with treatment outcomes [94,95,96]. While this may seem like an appropriate task, because it is administered 4 weeks after drug administration and scan, any associated imaging findings involving cognitive control circuitry can only be indirectly inferred. To more directly probe the therapeutic effects of psilocybin in AUD, future imaging studies should involve participants in AA or AA/TSF-based treatments; utilize the AA-based reappraisal task described by Galanter [109] to investigate whether psilocybin is associated with increased activation in anterior MFG/area 9–46 during the “prayer” condition (i.e., directly probing cognitive control processes involved); and using PFM techniques, examine RSFC changes (1) within the DMN and (2) between anterior MFG/9–46 and the AI.

## 5. Conclusions

In this review, we have summarized the history and current state of the clinical trial literature on the therapeutic potential of psilocybin and other psychedelics in the treatment of AUD as well as the existing neuroimaging literature. Key pitfalls include the neglect of AA/TSF frameworks in current experimental designs; lack of appropriate measurements of the spiritual/mystical experiences that may be necessary for behavior change; lack of direct comparisons to effective, existing pharmacotherapeutic agents for AUD; and, regarding neuroimaging specifically, gaps in study design that fail to directly probe the cognitive control processes that may be involved in the therapeutic effect of psilocybin in the treatment of AUD.

These issues substantially limit the ability to interpret the current literature examining therapeutic effects of and mechanisms in psilocybin and other psychedelics in the treatment of AUD. To improve interpretability and reproducibility, future trials should incorporate objective primary outcomes (e.g., abstinence, sustained PHDD reduction, engagement in AA), preregistration of analytic plans, and independent endpoint adjudication. For mechanistic neuroimaging studies, clearly defining target circuits (e.g., anterior and posterior MFG, anterior insula, and their functional connectivity), relevant task paradigms, and consideration of appropriate acquisition parameters and sample size to ensure reliability will be essential. Linking these neural measures prospectively to alcohol use outcomes will allow direct tests of whether circuit-level engagement mediates recovery-related behavior. If future studies consider these recommendations, psychedelics can be appropriately evaluated and potentially offer hope for otherwise treatment-refractory patients.

## Figures and Tables

**Figure 1 brainsci-15-01190-f001:**
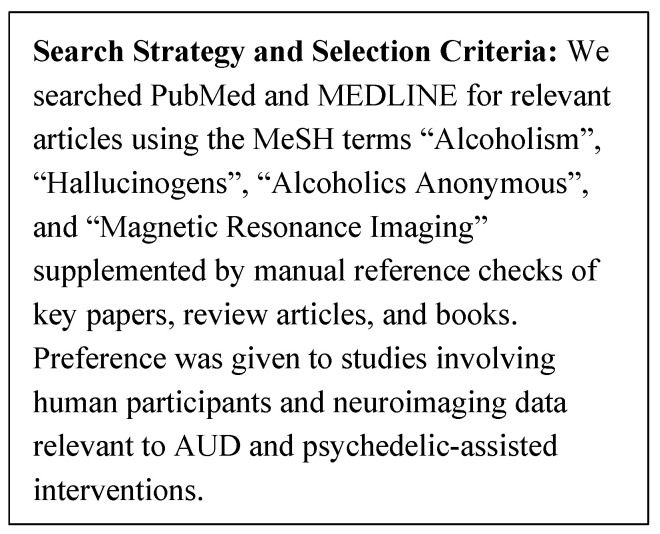
Search Strategy and Selection Criteria.

**Figure 2 brainsci-15-01190-f002:**
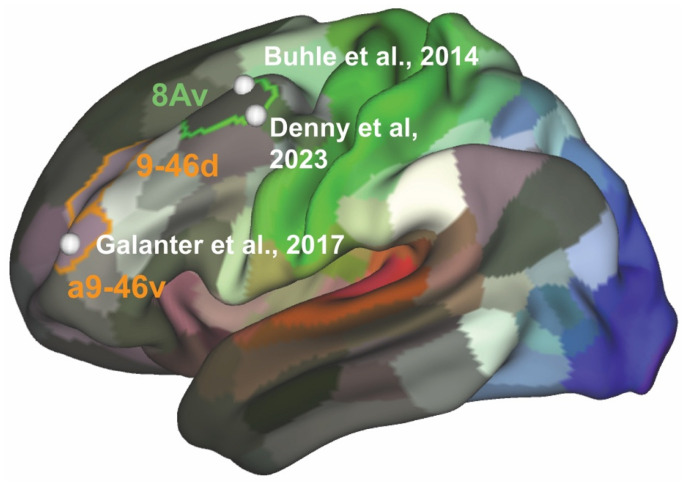
Meta-analytic peaks from reappraisal tasks [119,120] localize to a more posterior DLPFC area (green, area 8), whereas regulation-related brain activity in sober AA members [109] is centered in a more anterior area (orange, DLPFC area 9-46). These regions of DLPFC are shown on the Human Connectome Project Multimodal Parcellation [122], visualized in Connectome Workbench V 1.5 [121].

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
