# Peer review of "A Tragedy of Errors: The State of Psychedelic Research in the Treatment of Alcohol Use Disorder"

_brainsci, 2025, doi:10.3390/brainsci15111190_

Round 1
Reviewer 1 Report
Comments and Suggestions for Authors
This review aims to summarize the history and current state of clinical trials on psilocybin and psychedelics for AUD, along with neuroimaging literature. The authors emphasize that the main issues limiting the interpretation of the available literature include neglecting AA/TSF in experimental designs, lacking measurements of experiences necessary for behavior change, the absence of direct comparisons to effective pharmacotherapies, and neuroimaging gaps that do not directly assess cognitive control processes involved in psilocybin's therapeutic effects.
The study efficiently assessed the issue, and some points that may clarify the current status are provided in the attached PDF file, highlighted in blue.

Author Response
Response to Reviewer 1
We thank the reviewer 1 for the overall supportive comments. We have addressed each comment as follows:
- Please add the abbreviations used in this review.
- Line 391and 444: Abbreviations for dorsolateral prefrontal cortex
We have removed “dorsolateral prefrontal cortex” from line 344 given the abbreviation is spelled out previously in 391 - BOLD (line 400)
- Line 391and 444: Abbreviations for dorsolateral prefrontal cortex
We have added “blood-oxygenation-level-dependent (BOLD)” to clarify the meaning
- Line 408: As mentioned earlier (line 398), a similar limitation related to sample size might also apply to this study.
We have clarified that PFM techniques allow for reliable analyses using small sample sizes.
- Please provide a higher-resolution map and allocate more space in the manuscript
to help readers better understand the visualization.
- We have included a higher resolution brain map
- We have revised the caption as follows: “Figure 1: Meta-analytic peaks from reappraisal tasks [119,120] localize to a more posterior DLPFC area (green, area 8), whereas regulation-related brain activity in sober AA members [109] are centered in a more anterior area (orange, DLPFC area 9 -46). These regions of DLPFC are shown on the Human Connectome Project Multimodal Parcellation[135], visualized in Connectome Workbench V 1.5[136] .”
- In the manuscript we have expanded on the description of the figure: “Figure 1 visualizes this distinction, underscoring that “regulation” is not a unitary construct: ROC-based reappraisal and AA-related recovery processes engage spatially and functionally distinct DLPFC subregions.”
- References 135 and 136 in the figure
We have rearranged the references to correspond to the order in which they appear in the text - If possible, please include a table summarizing the studies listed at https://clinicaltrials.gov/search?cond=Alcohol%20Use%20Disorder&intr=Psilocybin&page=
We appreciate this suggestion. However, given that the ongoing and registered psilocybin-AUD trials have been comprehensively reviewed elsewhere (Wittenkeller et al., 2025, “Psychedelics as pharmacotherapeutics for substance use disorders: A scoping review on clinical trials and perspectives on underlying neurobiology”), we have opted not to reproduce that material here. Our focus in this article is on the conceptual and mechanistic issues underlying the interpretation of completed studies, rather than cataloguing trial parameters of currently ongoing and registered studies. We have cited this recent review at Reference 47 to direct interested readers to a complete summary.
We have also pointed readers towards this review in the manuscript: “These results were heralded as extremely promising [46], prompting additional trials investigating the use of psilocybin assisted psychotherapy in the treatment of AUD, the details of which are reviewed elsewhere[47]”
- Reference formatting
We have corrected the references highlighted as follows:
- Kurtz, E. Not-God: A History of Alcoholics Anonymous; Hazelden: Center City, MN, 1991.
- James, W. The Varieties of Religious Experience; Touchstone.; Simon and Schuster, Inc: New York, NY, 2004.
- The Use of LSD in Psychotherapy and Alcoholism; Abramson, H.A., Ed.; The Bobbs-Merrill Company, Inc, 1967.
- Hoffer, A.; Osmond, H. The Hallucinogens; Academic Press: New York, 1967.
- Guyatt, G.; Rennie, D.; Meade, M.O.; Cook, D.J. Users’ Guides to the Medical Literature: A Manual for Evidence-Based Clinical Practice, 3rd Ed; McGraw-Hill Education, 2015.
- Begotti, T.; Borca, G.; Rabaglietti, E.; Ciairano, S. Fattori Associati All’interruzione Del Consumo Di Sostanze Psicoattive in Adolescenza: Uno Studio Sui Valori, l’uso Del Tempo Libero, i Modelli e Gli Atteggiamenti Dei Genitori. Psicologia Clinica dello Sviluppo 2, 427–447.
- Siegel, J.S.; Subramanian, S.; Perry, D.; Kay, B.P.; Gordon, E.M.; Laumann, T.O.; Reneau, T.R.; Metcalf, N.V.; Chacko, R.V.; Gratton, C.; et al. Psilocybin Desynchronizes the Human Brain. Nature, 632, 131-138, 2024, doi:10.1038/s41586-024-07624-5.

Reviewer 2 Report
Comments and Suggestions for Authors
Hello, the article is excellent. To improve it, please correct the following. The result should be stated more clearly in the abstract. The methodology of the article is not discussed. Please state it in full. The mechanism of the disease and how psychedelics work in these pathways should be discussed. You can use the following article: https://link.springer.com/article/10.1007/s12035-025-05097-9. Animal and cellular data should be added. A mechanism should be presented.
Author Response
Response to Reviewer 2
We thank Reviewer 2 for the helpful comments and feedback. We have made the following changes
- The result should be stated more clearly in the abstract.
We revised the abstract to include a clearer statement specifying the limitations of current mechanistic neuroimaging studies:
“Additionally, current mechanistic neuroimaging studies examining the therapeutic effects of psychedelics in AUD are limited by design and do not directly interrogate the cognitive and circuit-level processes likely underlying treatment response.”
- The methodology of the article is not discussed. Please state it in full.
Because this is a narrative review, we did not include a forma “Methods” section. However, we have added a search strategy:
“Search Strategy and Selection Criteria: We searched PubMed and MEDLINE for relevant articles using the MeSH terms “Alcoholism”, “Hallucinogens”, “Alcoholics Anonymous”, and “Magnetic Resonance Imaging” supplemented by manual reference checks of key papers, review articles, and books. Preference was given to studies involving human participants and neuroimaging data relevant to AUD and psychedelic-assisted interventions.”
- The mechanism of the disease and how psychedelics work in these pathways should be discussed. You can use the following article: https://link.springer.com/article/10.1007/s12035-025-05097-9. Animal and cellular data should be added. A mechanism should be presented.
We have clarified the discussion of mechanistic models in the imaging section, noting that while receptor-level pharmacology (particularly 5-HT2A receptor activity) has been well characterized, such pharmacodynamic explanations alone are insufficient to account for the observed therapeutic effects. We have therefore added text highlighting that complementary approaches—particularly those examining large-scale brain-network function via fMRI—may better capture the relevant neural processes underlying treatment response. This section describes a proposed, network-level mechanism in detail. To acknowledge the reviewer’s recommendation, we have also cited Lashgari et al. (2025) [Ref. 92] in this section.

Reviewer 3 Report
Comments and Suggestions for Authors
Please see the attachment.

Author Response
Response to Reviewer 3
We thank Reviewer for the very helpful, supportive, and thoughtful comments, and we have addressed them as follows.
- Add a brief note on how the literature was gathered, which bodies of work were prioritized, core databases and keywords, dates covered, and any language or study-type limits. Keep it short and narrative, not PRISMA…
We have added a search strategy:
“Search Strategy and Selection Criteria: We searched PubMed and MEDLINE for relevant articles using the MeSH terms “Alcoholism”, “Hallucinogens”, “Alcoholics Anonymous”, and “Magnetic Resonance Imaging” supplemented by manual reference checks of key papers, review articles, and books. Preference was given to studies involving human participants and neuroimaging data relevant to AUD and psychedelic-assisted interventions.”
- Calibrate claims to evidence strength by naming study design and giving one anchor number for key papers, and use steady wording to signal strong, mixed, or early evidence. I suggest to include some tables with studies with large cohorts
We appreciate the reviewer’s emphasis on clarity regarding evidence strength. Throughout the revised manuscript, we have explicitly identified study designs (e.g., open-label, randomized controlled, meta-analytic) and have included quantitative indicators—such as sample sizes, trial durations, and effect sizes—for the principal studies discussed (e.g., Hedges g = 0.52). We believe these details provide the requested “anchor numbers” and appropriately calibrate the strength of evidence presented.
We have included a table of the principle studies examining the efficacy of psychedelics in the treatment of AUD (Table 1)
- Separate the main threads so each has its own short section: clinical outcomes, expectancy and blinding, therapy standardization, and mechanistic imaging, with one line linking how each informs the next.
We appreciate the reviewer’s suggestion to clarify the flow between major conceptual domains. In the revised manuscript, we have reorganized the discussion of the 2022 Bogenschutz RCT so that limitations (functional unblinding, abstinence outcomes, and the absence of AA framing) follow immediately after the study summary, providing a natural transition to the subsequent sections. We also added Table 1, summarizing key quantitative results across major studies, which delineates the clinical outcomes thread. The following sections then elaborate on functional unblinding, abstinence as an outcome definition, and consideration of AA in study design, setting the stage for later discussion of mechanistic imaging. This reorganization achieves the intended clarity while maintaining the integrated narrative structure of the review.
- Add two or three high yield visuals: an evidence landscape mapping key studies by design, comparator, follow up, and primary endpoint, a one page comparison table of the most influential trials with outcomes and co interventions, and a simple conceptual figure linking the proposed mechanisms to the clinical outcomes discussed.
We thank the reviewer for these thoughtful suggestions. We have added Table 1, which summarizes key quantitative results from the most influential clinical trials, including study design, comparator, follow-up duration, and primary outcomes—addressing the reviewer’s recommendation for an evidence-landscape overview. We have also retained and clarified Figure 1, which provides the requested conceptual visualization linking circuit-level mechanisms to clinical outcomes. We believe these visuals together achieve the intent of the reviewer’s feedback, while avoiding redundancy and maintaining focus on the paper’s central conceptual aims.
- Make the “what next” guidance concrete: for open label designs specify objective primary outcomes, blinded assessment where possible, preregistration, and endpoint adjudication, and for imaging name target circuits, tasks, primary readouts, basic power assumptions, and how those map back to alcohol outcomes.
We have added the following to the “Conclusion” paragraph:
“To improve interpretability and reproducibility, future open-label and randomized trials should incorporate objective primary outcomes (e.g., abstinence, sustained PHDD reduction, engagement in AA), preregistration of analytic plans, and independent endpoint adjudication. For mechanistic neuroimaging studies, clearly defining target circuits (e.g., anterior and posterior MFG, anterior insula, and their functional connectivity), relevant task paradigms, and consideration of appropriate acquisition parameters and sample size to ensure reliability will be essential. Linking these neural measures prospectively to alcohol-use outcomes will allow direct tests of whether circuit-level engagement mediates recovery-related behavior.”
- Keep the tone balanced by flagging the most important null or mixed findings and stating what new data would most change confidence in the key claims.
We agree with the reviewer that balanced discussion is essential. Throughout the manuscript, we have emphasized the limitations, mixed findings, and methodological weaknesses of existing trials—including null effects on abstinence (e.g., Reiser et al., 2025), the limited durability of effects in the 2022 Bogenschutz RCT, and the interpretive constraints imposed by functional unblinding and study measurements in clinical trials and sample size, data acquisition methods, and task selection in neuroimaging studies. The paper explicitly frames these as reasons for caution and as motivation for future, appropriately powered and mechanistically targeted studies. Because these caveats are already embedded throughout each section, we believe the tone and balance of the manuscript appropriately reflect the current state of evidence.
- The English could be improved to more clearly express the research.
We appreciate the reviewer’s feedback regarding language quality. The manuscript has been carefully proofread to correct minor typographical and grammatical errors noted during revision, and we have also incorporated the reviewer’s suggestions to improve flow. We believe the English now clearly and accurately conveys the research content and meets the journal’s standards for clarity and readability.
